# Automatic Implementation Algorithm of Pressure Relief Drilling Depth Based on an Innovative Monitoring-While-Drilling Method

**DOI:** 10.3390/s22093234

**Published:** 2022-04-22

**Authors:** Zheng Wu, Wen-Long Zhang, Chen Li

**Affiliations:** School of Energy and Mining Engineering, China University of Mining and Technology (Beijing), Beijing 100083, China; bqt1900101036@student.cumtb.edu.cn (Z.W.); 13120008810@163.com (C.L.)

**Keywords:** vibration signals, neural network, drilling state identification algorithm, drilling depth, monitoring-while-drilling method

## Abstract

An innovative monitoring-while-drilling method of pressure relief drilling was proposed in a previous study, and the periodic appearance of amplitude concentrated enlargement zone in vibration signals can represent the drilling depth. However, there is a lack of a high accuracy model to automatically identify the amplitude concentrated enlargement zone. So, in this study, a neural network model is put forward based on single-sensor and multi-sensor prediction results. The neural network model consists of one Deep Neural Network (DNN) and four Long Short-Term Memory (LSTM) networks. The accuracy is only 92.72% when only using single-sensor data for identification, while the proposed multiple neural network model could improve the accuracy to being greater than 97.00%. In addition, an optimization method was supplemented to eliminate some misjudgment due to data anomalies, which improved the final accuracy to the level of manual recognition. Finally, the research results solved the difficult problem of identifying the amplitude concentrated enlargement zone and provided the foundation for automatically identifying the drilling depth.

## 1. Introduction

Vibration signals are often used in sensor monitoring [1,2,3], defect diagnosis [4,5,6,7] and engineering applications, such as gearboxes [8,9,10], aero-engines [11,12] and wind turbines [13]. Vibration signals in underground coal mining are often used to help the analysis of mining dynamic disasters, stress environments, drilling process and the state of surrounding rocks [14,15,16,17,18,19]. Microseismic (MS) monitoring, aimed at monitoring vibration signals, can detect the dynamic event of the surrounding rock and predict the rock burst disaster. Acoustic emission (AE) technology is widely used in the field of geotechnical engineering [20,21,22]. The AE signal contains spatial information about the complex structural distribution inside the material [23,24] and vast key information about the rock fracture evolution process [25]. The AE tomography can detect early internal damages, faults and abnormal regions with the distribution of velocity field in the drilling process [26,27]. A monitoring-while-drilling (MWD) system can employ the drilling signals to monitor the quality of borehole constructions, obtain the information of surrounding rocks and other useful information during the drilling process [28,29].

MS monitoring has gradually become the most conventional means of coal mine vibration signal analysis and is now widely used in China’s underground coal mines [30,31,32,33]. The mature technology of MS monitoring has solved a series of problems for actual underground coal mining. Similarly, drilling construction is an indispensable process for coal mine production, such as providing support, pressure relief and other required drilling engineering. Pressure relief drilling (PRD) is often used to reduce stress concentrations and avoid dynamic hazard events, such as rock bursts [34,35]; signal analysis during the drilling process can provide guidance for the drilling quality analysis and construction process. In the previous studies, Zhou et al. [36] proposed a hybrid rock recognition approach that combined Gaussian process regression with clustering, and employed MWD data and the adjusted penetration rate to achieve automated rock recognition. Liu et al. [37] analyzed the relationship between the transverse, longitudinal and torsional vibration of the drill rod and the properties of the rock being drilled. Zhang et al. [38] studied the drilling amplitude signals collected by the MS equipment, which can determine the drilling difficulty areas, sticking drilling and vibration events. Pu et al. [39] investigated the performance of ten frequently used machine learning models for MS/blasting event recognition. Faradonbeh et al. [40] discussed the applicability of three data mining techniques along with five conventional criteria to predict the occurrence of rock bursts in a binary condition.

The study results of [38] (Figure 1) pointed out that the periodic appearance of amplitude clusters can represent the drilling depth, which was of great significance to the automatic statistics of drilling depth and the supervision of workload. However, there is a lack of a method that can automatically and accurately identify amplitude clusters in the entire process of borehole construction. There are misjudgments caused by human subjective consciousness as only relying on the manual identification of amplitude clusters, which is inefficient and cannot meet the requirements of the efficient and safe operation of the mine.

In this study, a deep learning method is used to analyze the drilling amplitude signals collected by MS monitoring equipment. A fusion neural network is obtained based on Deep Neural Network (DNN) [41,42,43,44] and Long Short-Term Memory (LSTM) [45,46,47,48] algorithms, which can automatically distinguish and determine drilling information, such as the amplitude clusters, drilling start point, termination point and drilling duration of each section in an efficient and accurate manner.

The proposed rig drilling status identification algorithm can efficiently and accurately identify and analyze drilling operations, such as pressure relief drilling, support drilling in the underground coal mine, and obtain the actual construction length, construction sequence, time spent on each process and other details of the construction operations, as well as additional information, such as drilling difficulty and abnormal vibration information. It has far-reaching significance to ensure the quality and safe operation of underground construction and obtaining the properties of the roadway surrounding rocks.

The main contributions are as follows:(1)Divide all data into the training set, validation set and test set. The test set is not involved in the training and tuning of the neural network and is only used as the data for the final model effect evaluation to avoid the problem of information leakage that leads to the fake high identification accuracy of the neural network. The training and validation sets are divided by the stratified K-fold cross-validation method to find the optimal hyperparameters in the model training and tuning, which eliminates the influence of the imbalanced amount of data between the two categories on the model.(2)An efficient, automatic and precise neural network model is proposed to identify the drilling status of drilling rigs by drilling amplitude signals, which can fuse the data from single and multiple sensors, and the identification results from different neural networks.(3)An optimization method is presented, which is similar to “submerge” for two types of recognition anomalies caused by data in drilling state recognition by the neural network identification algorithm.

The remainder of this paper is organized as follows. In Section 2, we describe where and how we collected the drilling signal data and introduce basic concepts related to the neural network algorithm that is used in this paper. In Section 3, we demonstrate the composition and division of the dataset of the neural network in this paper, the preprocessing of the data and the structure of the proposed neural network recognition algorithm. In Section 4, we present and analyze the recognition results of the proposed algorithm, perform recognition error analysis, propose an error handling method and show the final recognition results. Finally, conclusions and future works are provided in Section 5.

## 2. Research Methods

### 2.1. Data Collection Method

MS monitoring equipment was selected to monitor the drilling process of PRD boreholes, which were located in three different underground coal mines in the Shandong Province, China. The PRD boreholes were drilled by a CMS1-6500/75 drill rig (shown in Figure 2a) with a length of 1 m per drill rod, and a new drill rod was added after each rod was drilled. The PRD boreholes were located at the side of the roadway, 1.5 m away from the roadway floor, with a drilling diameter of 150 mm and a design drilling depth of 30 m.

The MS monitoring system was arranged on one side of the PRD borehole, and contained three sensors, which are arranged as shown in Figure 2b. The three sensors are at different distances with the same PRD drilling hole, and the amplitudes of the drilling signals collected for the same drilling hole are different. When the drilling position changes, it cannot be guaranteed that a certain sensor always collects the maximum, minimum or middle amplitudes. In order to eliminate the influence of the distance from the borehole, the average value of the vibration amplitude signals collected by the three sensors was calculated, so that the amplitude signals of Sensor #1, Sensor #2 and Sensor #3 would be arranged from the smallest to the largest.

### 2.2. Neural Network Algorithm

For the problem of identifying the drilling status of the drilling rig, the collected drilling signals were learned and trained with the help of the excellent judgment accuracy of the deep neural network. For the identification of the drilling state of the drilling rig, it can be simplified as a binary classification [49]: the signal points in the drilling state as 1 (positive sample) and the signal points in extension of the drill rod, the non-drilling state, as 0 (negative sample). The middle layers of the neural network use a Rectified Linear Unit (ReLU) as the activation function, and the last layer uses a sigmoid activation function to output a probability value in the range of 0 to 1. The ReLU function resets all negative values to zero, while the sigmoid function “compresses” any value to the interval [0, 1], and its output value can be regarded as a probability value; the expressions of the two activation functions are given in Equations (1) and (2). Therefore, the network uses a binary cross-entropy loss function to calculate the loss, as in Equation (3).
(1)$$\mathrm{ReLU}(x)=\{\begin{array}{ll} x & if\;x>0\\ 0 & if\;x\leq 0 \end{array}$$
(2)$$S(x)=\frac{1}{1+e^{-x}}$$
(3)$$loss=-\frac{1}{N}\left[\sum_{i=1}^{N}y_{i}\cdot \log(p(y_{i}))+(1-y_{i})\cdot \log(1-p(y_{i}))\right]$$
where *N* represents the number of samples; *y_i_* represents the label value of sample *i*; and *p*(*y_i_*) represents the predicted probability value of the label value of sample *i*.

The practice of training and testing the model on all datasets is problematic, which can lead to the rapid over-fitting of the model on that dataset. Therefore, we divided the dataset into training set, verification set and test set in order to obtain a generalized model. The model was trained and learned on the training set, and the hyperparameters of the model were adjusted using the performance of the model on the validation set as a feedback signal. However, this causes information leakage when the model parameters are tuned multiple times on the validation set, and the model quickly over-fits on the validation set. So, we created a completely unused dataset, a test set, to evaluate the model, and the best parameters were determined by grid search [50] techniques. Then, the optimal parameters were used to re-train the model on all the training sets, and the effect of the model was finally evaluated on the test set.

Depending on the number of data points and the way the validation set is divided, it may result in a large variance in the validation scores, which makes it impossible to evaluate the model reliably. In this case, the best practice is to use the K-fold cross-validation [51] shown in Figure 3. This method divides the available data into K folds, instantiates K identical models, trains each model on K-1 folds and evaluates it on the remaining one. The validation score of the model is equal to the average of the K validation scores.

However, some classification problems may also show a large imbalance in the distribution of target classes, for example, there may be several times more negative samples than positive ones. In such cases, stratified sampling is used to ensure that the relative class frequencies are approximately the same in each training and validation fold. Stratified K-fold cross-validation [52] (Figure 4) is a variant of K-fold cross-validation, which returns stratified folds, with each fold containing roughly the same percentage of samples for each target category as the entire collection.

## 3. Design of Experiment

### 3.1. Composition of Experimental Data

We carried out drilling signal data acquisition in three coal mines (marked as mine A, B and C) in the Shandong Province, China. For mines A and B, the drilling signal data of one borehole was collected in each mine, marked as A1, B1. Additionally, the drilling signal data of five boreholes were collected in mine C, marked as C1~C5, with seven boreholes drilling rig amplitude data in total. In order to train and obtain a generalized neural network model, the amplitude data of one borehole A1 in mine A and two boreholes (C1, C3) in mine C were used as training data. The designed neural network was trained and verified by stratified K-fold cross-validation, and the data of one borehole B1 in B mine and three other holes (C2, C4, C5) in C mine are used as the test dataset to test the identification accuracy of the final model.

### 3.2. Pre-Processing of Experimental Data

The PRD drilling amplitude data from the training and validation sets were sorted, and the vibration signals within the drilling time were filtered according to the actual drilling time. The data points were manually labelled as 0 or 1 according to the time corresponding to the drilling state of the rig recorded during the field construction and the size of the drilling signal amplitude value (1 is the drilling state, 0 is the state of connecting the drill rod). The collected raw vibration signal data are used as raw data, as shown in Figure 5.

It can be easily seen from the Figure 5 that the collected vibration signal data contain some points with abnormally large amplitude values, which seriously deviate from the range of other amplitude value distributions; these outliers are randomly present in two different drilling states and the locations of outliers collected by different sensors may be different from each other. If these outliers are retained as inputs to the model, they have a great disturbance and impact on the subsequent training of the model and the accuracy of the final model; thus, it is necessary to remove the outliers from the input data before the model training process to avoid the model learning the wrong information and to ensure that the training and accuracy of the model are not affected by the outliers. Therefore, the 3σ principle [53,54] was used to filter the raw data in order to eliminate the influence of outliers on the model, that is, the (μ − 3σ, μ + 3σ) in each set of datasets is taken as the screening criterion for the outlier data. For the vibration signal data collected by each sensor, the abnormal value points exceeding 3σ are removed, which is shown in Figure 6.

The distribution of amplitude cluster and intervals can be seen clearly after removing the outliers, and the purpose of removing outliers and revealing the characteristics of the data was preliminarily achieved. To determine whether the two types of labels of the data are distinguishable in terms of the vibration amplitude index, the vibration signals data were represented according to the label classification as shown in Figure 7. For Sensor #1, the average amplitude of the drilling state is 1.85 and the average amplitude of the connecting state is 0.38. For Sensor #2, the average amplitude of the drilling state is 3.04 and the average amplitude of the connecting state is 0.47. For Sensor #3, the average amplitude of the drilling state is 9.87 and the average amplitude of the connecting state is 3.63. The vibration signals of the two states are well differentiated in terms of amplitude mean and maximum amplitude, and the difference features can be trained and learned by the designed neural network. Since the raw data are a combination of drilling amplitude values from three different boreholes in two different mines, there are some differences between the magnitude of amplitude values in different boreholes. In each individual borehole drilling data, the amplitude values of the two drilling states are still well separable. Therefore, the vibration amplitude can be used as a classification indicator to distinguish between the two drilling states.

Since the experimental data were divided into two categories and the number of data contained within the two categories was not equal, to make the model fully learn the characteristics of different types of data and improve its prediction accuracy, the stratified K-fold cross-validation method was used to avoid the model learning the characteristics only from one type of data, while the features of the other type are not sufficient learned. This study used stratified 10-fold stratified cross-validation. The training data were divided into training sets and validation sets in order to avoid information leakage caused by the model being adjusted directly on the test set during the learning process, which means that the entire training data were divided into 10 folds and the proportion of the two categories in each fold was approximately the same as the proportion in the total data set. The training was performed on 9 folds of the data each time, and the remaining 1 fold was used as the validation set to verify the model; the model effect was tested on a separate unused test set after the final model was trained.

### 3.3. Drilling State Identification Neural Network

The data, after the outlier removal and normalization process conducted in the previous section, were used as the input data of the neural network, and the neural network model was established using the LSTM and DNN methods. Three independent LSTM networks were built using single data from each of the three sensors as input data; one LSTM network and one DNN network were built using all the amplitude data collected by all three sensors as the input data. A total of five neural networks were established, and the architecture of the neural network is shown in Figure 8. Each neural network model was trained and validated separately, and the five neural networks jointly judged the drilling status of the rig using the respective vibration amplitudes and all vibration amplitudes collected by different sensors at the same moment. The entire process was constructed so that the amplitude data collected by the three sensors as a whole were input to the LSTM network and DNN network as input data, and the three different single sensor data were input to the LSTM networks #1, #2 and #3. The five sub-neural networks reveal the discrimination results of drilling state of drilling rig. For a certain moment, three drilling amplitude signals of the rig and five judgment results are obtained. Then, the three or more than three same drilling states were taken as the final drilling state result according to the majority rule.

The five sub-neural network models were trained and evaluated separately using the data in the training set as input data, and the feedback data (accuracy, Receiver Operating Characteristic and Area Under ROC Curve) obtained on the validation set were used to adjust and optimize the structure and parameters of the neural network (such as epochs, number of layers of deep neural network and number of neurons per layer). The training and adjustment were continued until the judgment performance of each neural network reached a good judgment accuracy rate. The final neural network model structure and parameters were determined after the comparison of the accuracy of model prediction results with different parameters.

After determining the final parameters, the training and validation sets were integrated into one training set. A new neural network was re-established according to the optimal neural network structure parameters and the training was restarted to ensure that each model could learn from the entire training set. That is, all the data were initially divided into training sets and verification sets to obtain the final neural network model, which was then applied to the data in the test set to finally evaluate the effect of the model.

## 4. Analysis and Discussion of the Experimental Results

### 4.1. Analysis of the Experimental Results

The four unused borehole data used as the test data were input into the final trained neural network model; the accuracy of the model is shown in Table 1 and the accuracy of each neural network in the model is shown in Table 2. The trained neural network models have a good identification accuracy, which are all over 97.00% and the average is 97.65%; their accuracy can effectively recognize the drilling state of the drilling signals collected in the field. The accuracy of the identification may not be satisfactory when only using single-sensor data for the identification, which is only about 92.72%, and the accuracy may become worse when encountering a more complex situation. Fusing the information of the recognition results of multiple sensors can effectively improve the identification accuracy of the final model and make the identification accuracy of the recognition model more robust.

### 4.2. Error Analysis

Comparing the identification results of the final model with the ground true, it was found that there are some discriminative abnormal points in the result. Taking part of the data in the Test #1 borehole as an example, and the partial discriminant abnormal point data shown in Table 3, we can observe that the data collected by the sensors corresponding to these distinguishing abnormal points are usually very different from the data of the surrounding points, which can also be seen in other datasets. Therefore, it can be assumed that the appearance of these discriminative anomalies has little to do with the discriminative neural network, but there are anomalies in the collected data.

The misjudgment of the identification results can be roughly divided into two categories. The first one is the “1110111” type of signals, that is, continuously or discontinuously sporadic signals in the continuous drilling state (1 state) are judged to be in the connecting state (0 state). In fact, the amplitude values of these points are usually very smaller, always one-half or one-third, than other points around them in a time series, so they can be regarded as anomalies. Combined with the actual situation on site, these points may be the sticking drill on-site. The other is the “0001000” type of signals, that is, sporadic signals mixed in the continuous connecting state (0 state) are judged as drilling state (1 state). Similarly, these points are abnormally different from other nearby points, usually 3 times or more higher than nearby points, and have a very short duration of 1 point with occasional cases lasting 2 points (8 s), so they can also be considered as anomalies. In the actual situation, these points may be the percussion made by the rig operator in order to lengthen the drill rod or faraway blasting, rock burst events or other strong vibration events that are collected by the sensor.

In order to eliminate the influence of the two types of anomalies mentioned above on the identification of drilling state, we propose an optimization method similar to “submerge”. The idea is that, when the state of a point is different from the state of the two points around it and the state of those two points is the same, the state of the point is modified to be the same as theirs. In addition, sometimes there are two consecutive points that are abnormal points (00011000, 11100111), but, in the actual drilling construction, the drilling state cannot only last for 8 s and the extension of the drill rod only takes 8 s. So, there is also a need for a separate “submerge” process for such consecutive abnormal points.

Based on this fact, we wrote a program to perform the “submerge” process of the 0 state abnormal points for each group of identification results, perform the “submerge” process of the 1 state abnormal points for the obtained results and process the points with consecutive outliers. In this way, we optimized the identification results and further improved the final accuracy of the discriminant. The final identification results after the “submerge” smoothing process are, respectively, shown in Figure 9, Figure 10, Figure 11 and Figure 12. The final identification case results of Test #1 are shown in detail, while the results of the other test groups are shown in abbreviated form. The final identification accuracy is almost the same as that of the manually labeled drilling state.

As shown in Figure 9, the blue and green rectangles in each graph are the identification results of the drilling status judged by the neural network algorithm. The length of each rectangle on the time axis is the time consumed by that section of the construction. The length of each drilling state is the length of one drill rod. Therefore, the length of each blue rectangle in the graph shows the time consumed by each drill rod, and thus indirectly indicates the drilling difficulty at that depth. The green rectangle between two adjacent blue rectangles is the non-drilling state, such as drill rod connection.

Test #1 is the borehole drilling amplitude data collected in mine C. Figure 9a–c shows the amplitude of the vibration signals collected by each sensor, while Figure 9d shows the amplitude signals of the three sensors fused into one graph. Since the three sensors were installed at different distances from the borehole, from Figure 9, we can see that the amplitude of the vibration signals collected by each sensor at the same time are different, but the overall trend is the same. It can be clearly seen that the drilling state identified by the neural network algorithm matches the area with high amplitude values due to the borehole construction, and accordingly, the connecting state matches the area with low amplitude values.

Test #2 is the borehole drilling amplitude data collected in mine B. Unlike Test #1, the source of this dataset, mine B, was not trained in the model. The model only learned the vibration amplitude data collected in mines A and C, which means that the model is completely unaware about the information related to mine B. As the four datasets used as the test set were not used in the model training and tuning process, there is no information leakage problem. This set of vibration signal data, from a completely new mine, has greater significance for the evaluation of model performance. This dataset eliminates the problem of the model having better identification accuracy in other boreholes in the same mine due to the knowledge of the mine’s geological conditions. The performance of the model on this dataset can better reflect the generalization ability of the model. As can be seen in Figure 10, the model has very good identification accuracy on this dataset, which comes from a brand new condition.

Test #3 and Test #4 are the other two borehole drilling amplitude data collected in mine C. It is also evident from Figure 11 and Figure 12 that the model still has a good recognition accuracy on these two test sets.

By using the above method, the state of the drilled hole, the amplitude clusters, intervals and the starting and ending points of drilling or drill rod connecting can be automatically and efficiently recognized with a high recognition accuracy. The vibration signals of downhole drilling can be identified and monitored continuously and efficiently. It can obtain, in a timely and accurate manner, the information of the whole process of the drilling construction, such as the construction depth of borehole drilling construction, construction time of each section of drill rod and the difficulty of construction at different depths. It can also ensure that the underground PRD boreholes are constructed according to the designed depth and the construction quality is effectively monitored, which is of great significance to ensure mine safety and obtain rock mass information.

## 5. Conclusions and Future Work

### 5.1. Conclusions

A neural network model for drilling rig drilling status identification that fuses single-sensor and multi-sensor prediction results was proposed in this study. The vibration signals of the drilling status of a same borehole were collected using multiple sensors, and the information of the identification results of single and multiple sensors were fused. The method was tested and verified; the identification accuracy of all four test datasets were over 97.00%, and the final identification accuracy was almost the same as that of the manually labeled drilling state after using the “submerge” optimization method. The results show that this method makes up for the deficiency of large errors (up to 6.32%) in the identification results due to the use of a single-sensor data source, and effectively improves the identification accuracy. The study is of great engineering significance to effectively identify and judge the construction length of underground borehole drilling and monitor the drilling information in the entire process of construction.

We innovatively proposed a drilling rig with a drilling status identification neural network algorithm that uses single-sensor and multi-sensor data and fuses multiple sub-neural network identification results. Several LSTM and DNN sub-neural networks were constructed using different drilling amplitude signal data sources. A new optimization method was proposed for the misjudgment caused by data anomalies in the identification results. It is a drilling state identification results optimization method that considers both the amplitude data of the time point and its neighboring points in the time dimension.

The main conclusions are as follows:(1)A high-accuracy neural network algorithm for the automatic identification of the drilling status of drilling rigs was proposed. The method uses single-sensor and multi-sensor data from the same borehole as input data and fuses the identification results from different types of sub-neural networks using different inputs, effectively improving the final identification accuracy. The identification accuracy of four test datasets of borehole amplitude data from two different mines were all above 97.00%.(2)An optimization method was proposed to deal with two types of misjudgment in the identification results due to data anomalies, and the optimized identification results are almost the same as the drilling status marked manually according to the actual construction status on-site.

### 5.2. Future Work

The underground environment is complex and there are a large number of noise sources, such as coal cutting by shearer loaders, blasting of heading face and overlying rock breaking. The noise signals are inevitably collected by the amplitude sensors, and are not only meaningless for the judgment of the drilling state of the drilling rig, but also affect the accuracy of model judgment. In addition, dynamic events, such as rock burst and coal and gas outburst, are completely eliminated in this study, which is very meaningful for research. Therefore, the next step of this study is to achieve the accuracy of noise elimination and the discriminative analysis of other dynamic events signals.

## Figures and Tables

**Figure 1 sensors-22-03234-f001:**
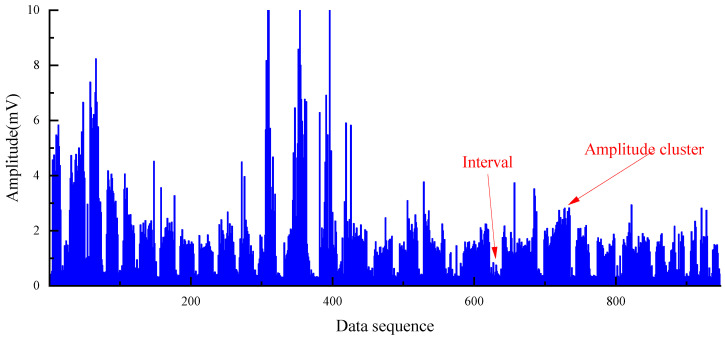
Amplitude clusters and intervals in amplitude data during drilling.

**Figure 2 sensors-22-03234-f002:**
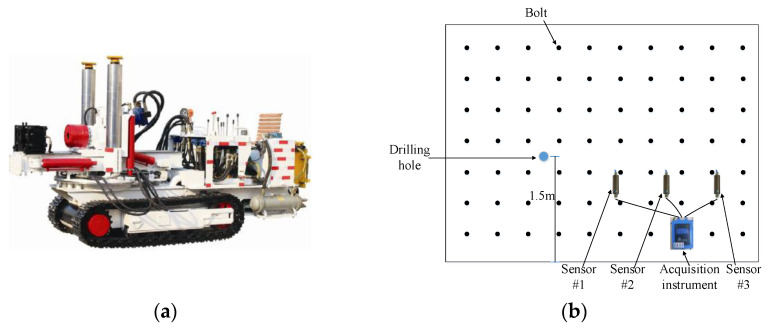
Drill rig and MS equipment layout. (**a**) CMS1-6500/75 drill rig; (**b**) Drill hole and MS equipment layout at roadway side.

**Figure 3 sensors-22-03234-f003:**
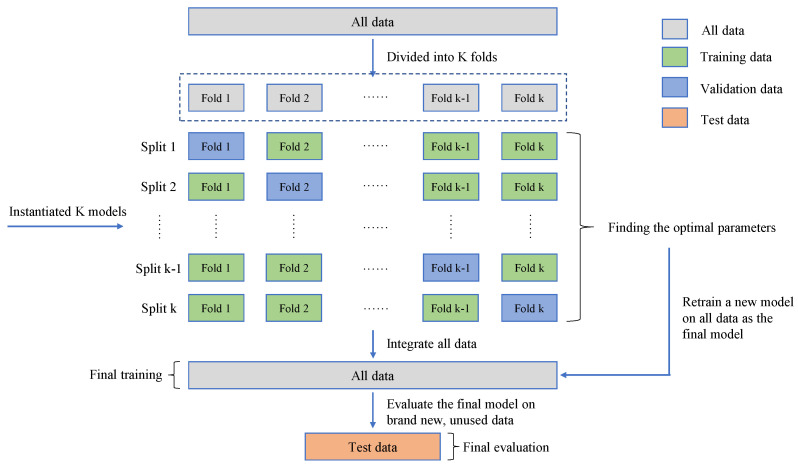
Schematic diagram of the K-fold cross-validation.

**Figure 4 sensors-22-03234-f004:**
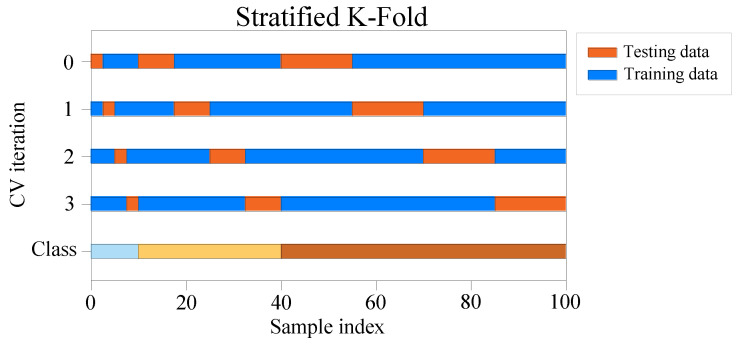
Schematic diagram of stratified K-fold cross-validation.

**Figure 5 sensors-22-03234-f005:**
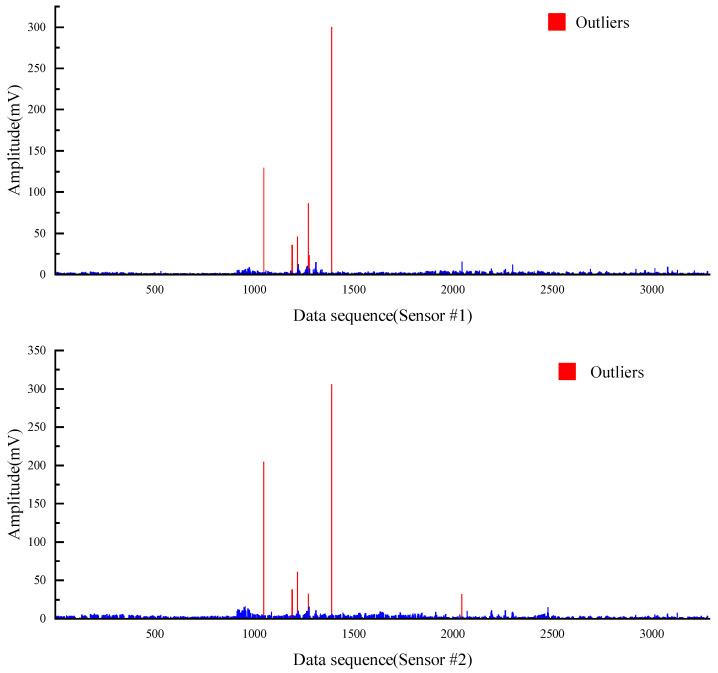
Original data of all 3 sensors.

**Figure 6 sensors-22-03234-f006:**
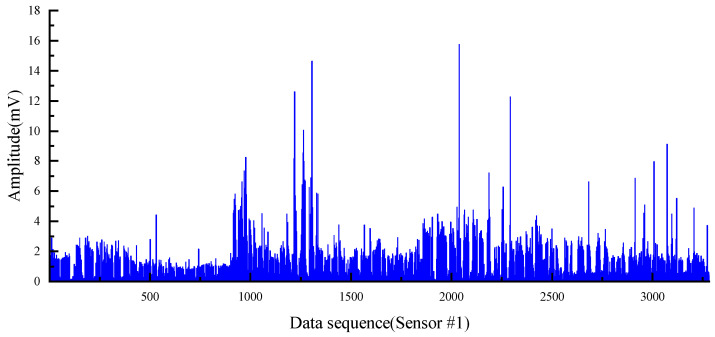
Amplitude signal data after removing the outliers.

**Figure 7 sensors-22-03234-f007:**
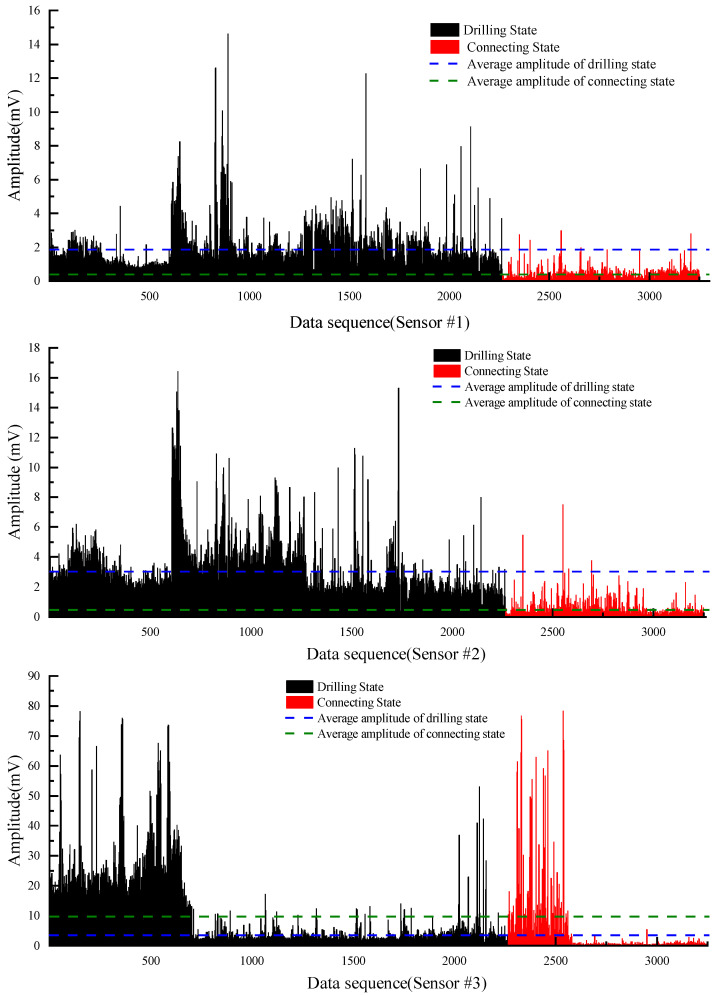
Separability of the two states.

**Figure 8 sensors-22-03234-f008:**
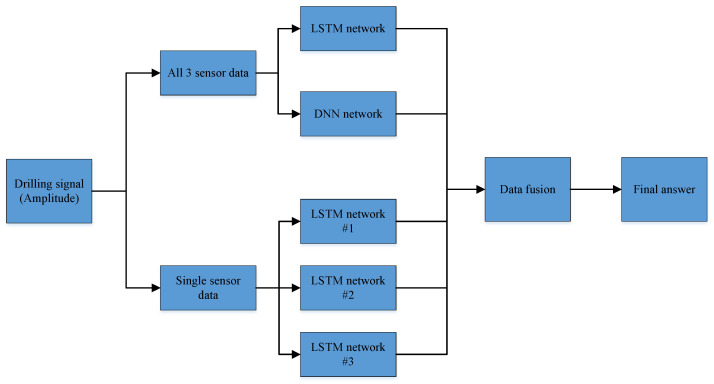
The architecture of the proposed identification neural network model.

**Figure 9 sensors-22-03234-f009:**
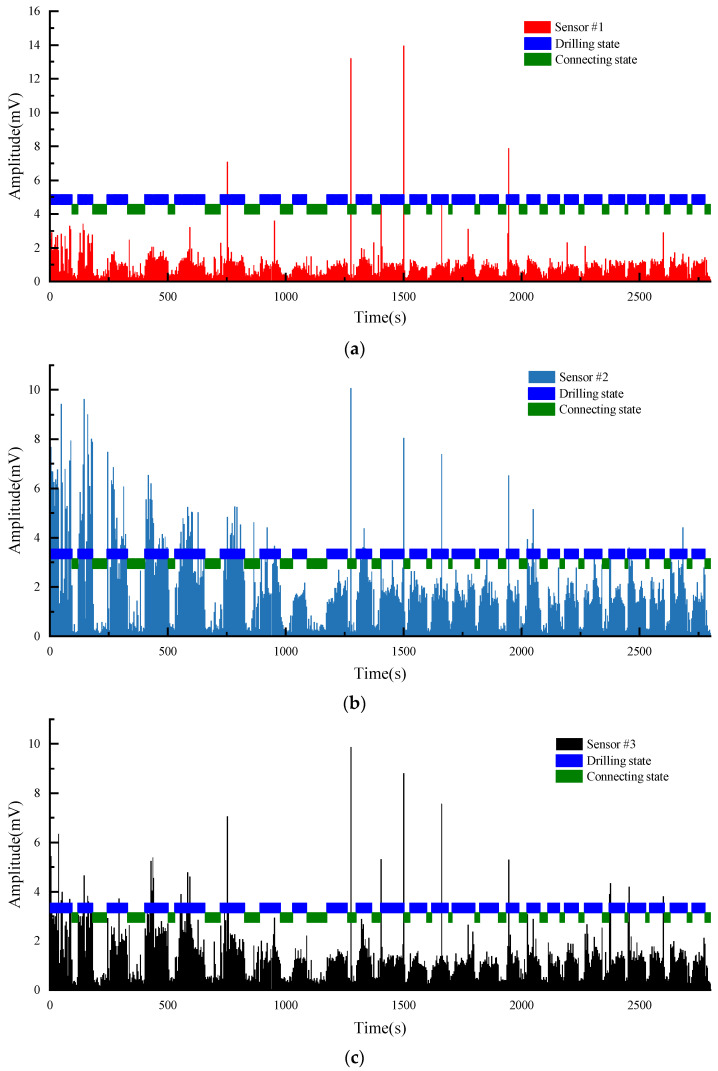
Test#1 drilling state final identification results. (**a**) Drilling state final identification result and vibration signal of Sensor #1. (**b**) Drilling state final identification result and vibration signal of Sensor #2. (**c**) Drilling state final identification result and vibration signal of Sensor #3. (**d**) Drilling state final identification result and vibration signal of all sensors.

**Figure 10 sensors-22-03234-f010:**
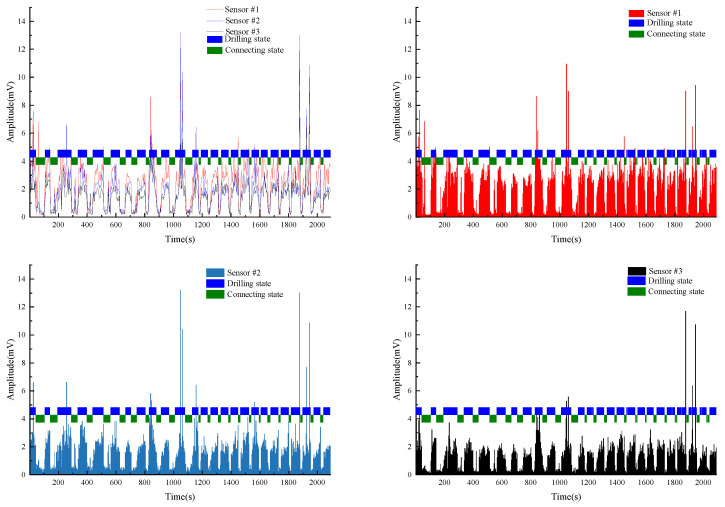
Test#2 drilling state final identification results.

**Figure 11 sensors-22-03234-f011:**
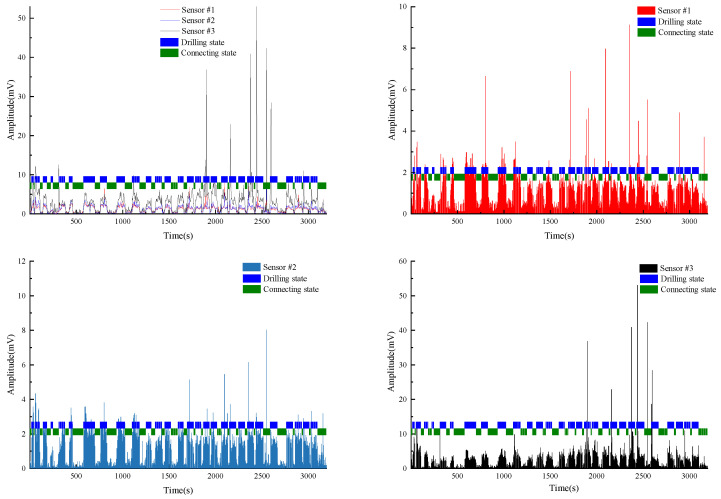
Test#3 drilling state final identification results.

**Figure 12 sensors-22-03234-f012:**
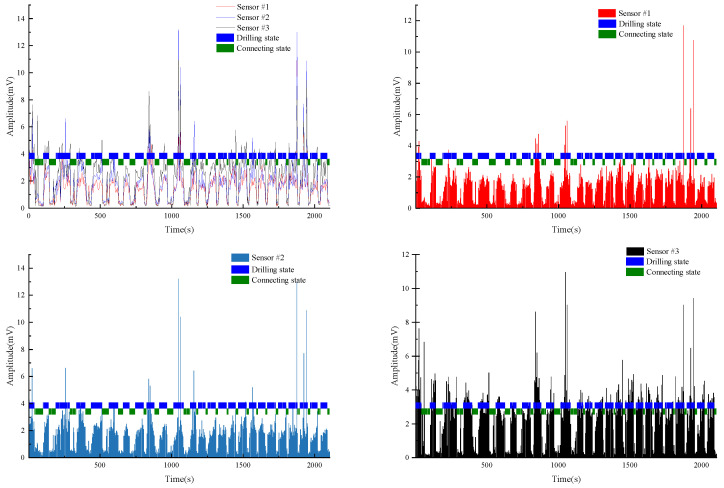
Test#4 drilling state final identification results.

**Table 1 sensors-22-03234-t001:** The accuracy of the neural network on the test data.

	Test Data #1	Test Data #2	Test Data #3	Test Data #4
Recognition accuracy	97.00%	98.47%	97.99%	97.15%

**Table 2 sensors-22-03234-t002:** The accuracy of each sub-neural network on the test data.

Network Type	Test Data #1	Test Data #2	Test Data #3	Test Data #4
LSTM #1	93.72%	92.72%	95.61%	97.72%
LSTM #2	95.15%	96.17%	97.37%	96.96%
LSTM #3	93.44%	92.15%	94.49%	96.20%
LSTM all	96.72%	93.10%	98.25%	96.96%
DNN	96.86%	98.47%	98.62%	97.15%

**Table 3 sensors-22-03234-t003:** Part of the outlier data points in the Test #1 dataset.

Data Sequence Number	Sensor #1 Amplitude	Sensor #2 Amplitude	Sensor #3 Amplitude	Drilling State Ground True	Drilling State Judged by the Network
⋯	⋯	⋯	⋯	⋯	⋯
17	2.707553	2.619514	6.796628	1	1
18	1.930003	2.793286	5.172647	1	1
19	1.963386	2.451638	5.28292	1	1
20	0.502925	0.59154	1.494293	1	0
21	1.324366	2.157405	4.256246	1	1
22	3.292366	3.701357	7.12655	1	1
23	3.088607	3.210542	7.949529	1	1
⋯	⋯	⋯	⋯	⋯	⋯
94	0.391242	0.756341	0.875585	0	0
95	0.225544	0.427893	0.178449	0	0
96	0.334536	0.450832	0.215356	0	0
97	1.083765	1.435536	2.665392	0	1
98	0.483574	0.586286	0.426804	0	0
99	0.425458	0.389063	0.18947	0	0
100	0.204848	0.269115	0.353566	0	0
⋯	⋯	⋯	⋯	⋯	⋯

## Data Availability

The data presented in this study are available upon request from the corresponding author.

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
