# Peer review of "Automatic Implementation Algorithm of Pressure Relief Drilling Depth Based on an Innovative Monitoring-While-Drilling Method"

_sensors, 2022, doi:10.3390/s22093234_

Round 1
Reviewer 1 Report
Dear Authors,
Based on the first round review of the manuscript entitled Automatic implementation algorithm of pressure relief drilling depth based on an innovative monitoring-while-drilling method, the reviewer has the following suggestions:
- In the introduction, the contribution is not clear. Please put a paragraph and explain deeply about the contribution in the revised manuscript.
- in the last paragraph of the introduction you should introducing the following sections in the manuscript. Please revised it and highlighted.
- How you can validate the performance in Neural Network algorithm in your work?
- The K fold cross validation is used to test the robustness, is it correct? How you can test the stability and reliability, as well?
- The authors needs to describe the main results in the conclusions.
- In paragraph “ Conclusions”, the novelty brought by the authors to solve the problems is not emphasized enough.
Regards,
Author Response
Point 1: In the introduction, the contribution is not clear. Please put a paragraph and explain deeply about the contribution in the revised manuscript.
Response 1:
First of all, we are very grateful to the reviewer for her/his good comments on the article. And we fully agree with the reviewer’s suggestions. We modify the Introduction of the article according to the reviewer’s suggestions.
Details are as follows:(Line 76 to 91)
“The main contributions are as follows:
- Divide all data into the training set, validation set, and test set. The test set is not involved in the training and tuning of the neural network and is only used as the data for the final model effect evaluation to avoid the problem of information leakage that leads to the fake high identification accuracy of the neural network. The training and validation sets are divided by the stratified K-fold cross-validation method to find the optimal hyperparameters in the model training and tuning, which eliminates the influence of the imbalance amount of data between the two categories on the model.
- We propose an efficient, automatic, and precise neural network algorithm to identify the drilling status of drilling rigs by drilling amplitude signals, which can fuse the data from single and multiple sensors, and the identification results from different neural networks.
- We present a smoothing optimization method similar to " Submerge" for two types of recognition anomalies caused by data in drilling state recognition by neural network identification algorithm.”
Point 2:In the last paragraph of the introduction you should introducing the following sections in the manuscript. Please revised it and highlighted.
Response 2:
Appreciate the comment. We have added the structure introduction to the Introduction paragraph. Details are as follows:(Line 92 to 100)
“The remainder of this paper is organized as follows.In Section 2, we describe where and how we collected the drilling signal data and introduce basic concepts related to the neural network algorithm that used in this paper. In Section 3, we demonstrate the composition and division of the data set of the neural network in this paper, the preprocessing of the data, and the structure of the proposed neural network recognition algorithm. In Section 4, We present and analyze the recognition results of the proposed algorithm, perform recognition error analysis, propose an error handling method, and show the final recognition results. Finally, conclusions and future works are provided in Section 5”
Point 3:How you can validate the performance in Neural Network algorithm in your work?
Response 3:
Thanks for the reviewer’s comment.Underfitting, overfitting, and the generalization ability of the modelare always been the most concerned issuefor the neural networkalgorithm.
In order to obtain a generalized model which not overfitting on some specific circumstances, and performwell on the datasets that the model has never seen or trained before.We collected rig amplitude data from a total of seven boreholes (denoted as A1, B1, C1~C5) in three different underground coal mines (denoted as A,B,C, respectively) in Shandong Province, China.Due to the use of supervised learning,all data are manually annotated according to the time corresponding to the drilling state of the rig recorded during the field construction.
The accuracy of the model prediction is the accuracy between the drilling state given by the model and the manually labeled drilling state.Since the drilling states are classified into two categories, the model training, and hyperparameter optimization processuses the binary cross-entropy loss function to calculate the loss, which is a common and standard loss function for the binary classification problem.
In order to train a generalized neural network model,the vibration data of one hole A1 in Mine A and two holes (C1, C3) in Mine C are used as training data,and the designed neural network was trained and validated by Stratified K-fold cross-validation.and the feedback data (accuracy, Receiver Operating Characteristic(ROC), and Area Under ROC Curve(AUC)) obtained on the validation set are used to adjust and optimize the structure and parameters of the neural network (such as epochs, number of layers of the deep neural network, number of neurons per layer, etc.).
One hole B1 in mine B and three other holes (C2, C4, C5) in mine C, a total of four borehole vibration signal data from two different mines,were used as the test dataset to test the discriminative accuracy of the final model obtained from the training.The accuracy of the neural network in predicting the drilling status of the rig in these four test sets was 97.00%, 98.47%, 97.99%, and 97.15%, respectively.These 4 sets of final test data were never used in the training and tuning of the model, so there is no information leakage problem.And the final testdatasets are from two different underground coal mines(B, C), which excludes the overfitting problem that the model has good prediction accuracy in a specific mine condition, but has poor prediction results under other mine conditions.Given the good prediction accuracy of the model on all four test datasets (all over 97%),we believe that the obtained model still performs well under brand new mine drilling data conditions, which validates the accuracy and validity of the obtained model.
Point 4:The K fold cross validation is used to test the robustness, is it correct? How you can test the stability and reliability, as well?
Response 4:
Thanks for the reviewer’s comment.A common and basic training and validation method for the neural network is hole-out validation.Set aside a certain percentage of data as the test set, train the model on the remaining data, and then evaluate the model on the test set.If the dataset is small, the divided validation set will be very small.Therefore, the validation score can fluctuate widely, depending on how the validation and training sets are divided.That is, the way the validation set is divided may cause a large variance in the validation scores, making it impossible to reliably evaluate the model.
The best practice, in this case, is to use K-fold cross-validation.This method divides the available data into K partitions, instantiates K identical models, train each model on K-1 partitions, and evaluate it on the remaining one.The validation score of the model is equal to the average of K validation scores. This method has been tested and has a good performance in the Kaggle competition.In the model training and tuning process,the prediction accuracy and the loss obtained by the binary cross-entropy loss function of the model under different folds, and different hyperparameters are concerned to determine the optimal parameters of the model.After determining the final parameters, the training and validation sets are integrated into one training set. Re-establish a new neural network according to the optimal neural network structure parameters obtained by the K-fold cross-validation and restart the training to obtain the final neural network model, which is then applied to the data in the test set to finally evaluate the effect of the model.
Point 5:The authors needs to describe the main results in the conclusions.
Response 5:
Many thanks for the valuable comment from the reviewer. We have modified the “Conclusion” part, details are as follows:(Line 378 to 388)
“The main conclusions are as follows:
1) A high-accuracy neural network algorithm for automatic identification of drilling status of drilling rigs is proposed. The method uses single-sensor and multi-sensor data from the same borehole as input and fuses the identification results from different types of sub-neural networks using different inputs, effectively improving the final identification accuracy. The identification accuracyof four test datasets of boreholeamplitude data from two different mines are all above 97.00%
2) An optimization method is proposed for two types of misjudgment in the identification results due to data anomalies, and the optimized identification results are almost the same as the drilling status marked manually according to the actual construction status on-site.”
Point 6:In paragraph “Conclusions”, the novelty brought by the authors to solve the problems is not emphasized enough.
Response 6:
Thanks for the comment. We have revised the “Conclusion” according to the reviewer’s suggestion, details are as follows:(Line 371 to 377)
“We innovatively proposed a drilling rig drilling status identification neural network algorithm which uses single-sensor and multi-sensor data and fuses multiple sub-neural network identification results.Several LSTM and DNN sub-neural networks were constructed using different drilling amplitude signaldata sources.A new optimization method is proposed for the misjudgment caused by data anomalies in the identification results, a drilling state identification results optimization method that considers both the amplitude data of the time point and its neighboring points in the time dimension.”
Reviewer 2 Report
Please find the comments referring to the paper as an attachment.

Round 2
Reviewer 1 Report
Dear Authors,
Thank you for your response letter. Based on the 2nd round review of the manuscript entitled Automatic implementation algorithm of pressure relief drilling depth based on an innovative monitoring-while-drilling method, it can be accepted for further processing.
Regards,
Author Response
Thank you for the positive feedback on our revised manuscript.
Reviewer 2 Report
All comments of the reviewer have been included in the revised version of the paper. I recommend publication this paper in its present form.
Author Response
Thank you for your positive feedback on our revised manuscript.